# Predictive Validity of the qSOFA Score for Sepsis in Adults with Community-Onset Staphylococcal Infection in Thailand

**DOI:** 10.3390/jcm8111908

**Published:** 2019-11-07

**Authors:** Supaksh Gupta, Kristina E. Rudd, Sarunporn Tandhavanant, Pornpan Suntornsut, Ploenchan Chetchotisakd, Derek C. Angus, Sharon J. Peacock, Narisara Chantratita, Timothy Eoin West

**Affiliations:** 1Department of Medicine, University of Washington, Seattle, WA 98195, USA; 2Division of Pulmonary, Sleep, and Critical Care Medicine, University of Washington, Seattle, WA 98195, USA; ruddk@pitt.edu (K.E.R.); tewest@uw.edu (T.E.W.); 3Department of Critical Care Medicine, University of Pittsburgh, Pittsburgh, PA 15260, USA; angusdc@ccm.upmc.edu; 4Department of Microbiology and Immunology, Faculty of Tropical Medicine, Mahidol University, 420/6 Rajvithi Road, Bangkok 10400, Thailand; sarunporn.tan@mahidol.ac.th; 5Mahidol-Oxford Tropical Medicine Research Unit, Faculty of Tropical Medicine, Mahidol University, Bangkok 10400, Thailand; pornpan@tropmedres.ac; 6Department of Medicine, Khon Kaen University, Khon Kaen 40002, Thailand; ploencha@kku.ac.th; 7Department of Medicine, University of Cambridge, Cambridge CB2 0QQ, UK; sjp97@medschl.cam.ac.uk

**Keywords:** sequential organ failure assessment scores, sepsis, systemic inflammatory response syndrome, Thailand, Staphylococcus

## Abstract

The quick sequential organ failure assessment (qSOFA) score has had limited validation in lower resource settings and was developed using data from high-income countries. We sought to evaluate the predictive validity of the qSOFA score for sepsis within a low- and middle-income country (LMIC) population with culture-proven staphylococcal infection. This was a secondary analysis of a prospective multicenter cohort in Thailand with culture-positive infection due to *Staphylococcus aureus* or *S. argenteus* within 24 h of admission and positive (≥2/4) systemic inflammatory response syndrome (SIRS) criteria. Primary exposure was maximum qSOFA score within 48 h of culture collection and primary outcome was mortality at 28 days. Baseline risk of mortality was determined using a multivariable logistic regression model with age, gender, and co-morbidities significantly associated with the outcome. Predictive validity was assessed by discrimination of mortality using area under the receiver operating characteristic (AUROC) curve compared to a model using baseline risk factors alone. Of 253 patients (mean age 54 years (SD 16)) included in the analysis, 23 (9.1%) died by 28 days after enrollment. Of those who died, 0 (0%) had a qSOFA score of 0, 8 (35%) had a score of 1, and 15 (65%) had a score ≥2. The AUROC of qSOFA plus baseline risk was significantly greater than for the baseline risk model alone (AUROC_qSOFA_ = 0.80 (95% CI, 0.70–0.89), AUROC_baseline_ = 0.62 (95% CI, 0.49–0.75); *p* < 0.001). Among adults admitted to four Thai hospitals with community-onset coagulase-positive staphylococcal infection and SIRS, the qSOFA score had good predictive validity for sepsis.

## 1. Introduction

Sepsis, a significant cause of morbidity and mortality worldwide, disproportionately affects individuals from low- and middle-income countries (LMICs) [1]. Although robust data on the global burden of sepsis are lacking, it is estimated that over 80% of the worldwide mortality from sepsis occurs in LMICs [2]. *Staphylococcus aureus* is one of the most frequent causes of serious infection in LMICs and is associated with an increased risk of death when compared to other bacterial pathogens [3,4,5,6]. *Staphylococcal argenteus*, a genetically divergent lineage from *S. aureus* that is indistinguishable from *S. aureus* using routine diagnostic microbiology methods, has also become a more frequently reported cause of bacteremia [7].

The consensus definition for sepsis has been updated several times to reflect contemporary understanding of sepsis biology [8,9,10,11]. In 2016, the Sepsis-3 Task Force met to update the definition of sepsis to better reflect the current pathobiologic understanding of sepsis and the overemphasis of the prior definition on inflammation [11]. The ‘quick’ SOFA (qSOFA) score was proposed by the Sepsis-3 Task Force as a tool to assist in early identification of patients at risk of sepsis [9]. The qSOFA score was created using retrospective statistical analysis of cohorts exclusively from high-income countries (HIC) [11]. To date, qSOFA has been primarily validated in HICs, with more limited validation in LMICs [12,13,14]. Furthermore, no studies have reported the performance of qSOFA among patients in LMICs with documented staphylococcal infection.

Therefore, the purpose of this study was to evaluate the predictive validity of qSOFA for sepsis within a LMIC population of hospitalized adults with community-onset staphylococcal infection and positive systemic inflammatory response syndrome (SIRS) criteria.

## 2. Experimental Section

This study was a secondary analysis of a prospective observational multicenter cohort study of hospitalized adult and adolescent patients with community-onset staphylococcal infection in Thailand, a middle-income country [15,16,17,18]. Subjects were recruited from four hospitals in northeast Thailand between March 2010 and December 2013. Inclusion criteria for patients in the original study were age ≥14 years old, admitted to an acute care ward or intensive care unit, born in Thailand, had a culture taken within 48 h of admission from any sterile site that was determined by the hospital microbiology laboratory to grow *S. aureus*, and met at least two of four SIRS criteria within 48 h of culture sample collection. SIRS criteria were defined as: body temperature > 38 degrees Celsius or <36 degrees Celsius; heart rate > 90 beats/minute; respiratory rate > 20 breaths/minute, PCO_2_ < 32 mmHg, or ventilator requirement; and white blood cell count (WBC) > 12,000 cells/mL, WBC < 4000 cells/mL, or band forms > 10% [8].

Patients were excluded from the study if they were pregnant, had recently received high doses of immunosuppressant medications, had received chemotherapy within the past 3 months, had history of chronic infection with pathogens such as tuberculosis or HIV, or if the cultures obtained from the sterile sites contained pathogens other than *S. aureus* (as determined by the hospital microbiology laboratory).

Enrolled patients’ lowest systolic blood pressure (SBP) and Glasgow Coma Scale (GCS) within 48 h after admission were recorded, along with co-morbidities, including but not limited to cardiac disease, renal disease, diabetes mellitus, autoimmune disease, liver disease, connective tissue disease, and cancer. Additional collected clinical data included laboratory test results, quantity of intravenous fluids administered within the first 48 h of admission, vasopressors received, ventilator requirements, receipt of surgical debridement or abscess drainage, and antibiotics administered. If discharged alive from the hospital, patients were contacted by phone to determine mortality at 28 days after study enrollment.

### 2.1. Bacterial Isolates

Three hundred and twenty-seven patients met enrollment criteria for the original study of which 311 had isolates available for further analysis. Subsequent analysis of the available isolates by pulsed field electrophoresis (PFGE) and multilocus sequence typing (MLST) identified 58/311 (19%) as *S. argenteus*, a genetically divergent lineage of *S. aureus* that is coagulase-positive [17].

### 2.2. Statistical Analysis

For this secondary analysis, patients enrolled within the original study were excluded if they were < 18 years old or if initial specimens for culture were obtained >24 h after hospital admission. This was to avoid including patients with hospital-acquired infections. Primary exposure was maximum qSOFA score within 48 h of specimen collection. The qSOFA score was calculated using altered mental status (defined as GCS ≤ 14), systolic blood pressure ≤ 100, and respiratory rate ≥ 22 [11]. For patients who were intubated, GCS verbal score was extrapolated from motor and eye scores [19]. Exact respiratory rate was not explicitly recorded in the data, only whether the rate exceeded 20 breaths/minute. Thus, patients were assigned one point for respiratory rate > 20.

The primary outcome was predictive validity of qSOFA for sepsis, using mortality at 28 days as a surrogate marker. A fixed time point for mortality was selected to reduce inter-hospital variability due to variations in discharge policies [20]. Predictive validity, a form of criterion validity, is used to assess conditions such as sepsis that do not have a gold standard test for diagnosis and thus cannot be determined with complete certainty [11]. A major feature of sepsis is the presence of “life-threatening organ dysfunction”, where individuals who develop sepsis are more likely to die compared to those with uncomplicated infections. Mortality at 28 days is therefore more likely to be associated with sepsis and thus was the outcome selected to evaluate the predictive validity of qSOFA for sepsis.

To assess predictive validity, a model of baseline risk of mortality was compared to a model of baseline risk plus the qSOFA score to evaluate the potential of the qSOFA score to identify those patients with excess risk of death above and beyond their baseline risk. Logistic regression was performed with all recorded co-morbidities to determine which of those were significantly associated with mortality at 28 days, of which liver disease and cardiac disease were significant. A baseline risk model for mortality was then developed using logistic regression, with 28-day mortality as the outcome and liver disease (present/absent), cardiac disease (present/absent), sex, and age in years as the exposure variables. To quantify predictive validity, the qSOFA score was added to the baseline risk model to determine excess mortality above and beyond the baseline risk model. The two models were compared using area under the receiver operating characteristic (AUROC) curves, with statistical significance defined as *p* < 0.05. Statistical analyses were conducted using Stata version 13.0 (College Station, TX, USA).

### 2.3. Ethics

Ethical approval was obtained from the following Ethical and Scientific Review committees: Faculty of Tropical Medicine, Mahidol University (approval no. MUTM 2011-007-01); Sunpasitthiprasong Hospital, Ubon Ratchathani (approval no. 004/2553); Udon Thani Hospital, Udon Thani (approval no. 0027.102/2349); Khon Kaen Hospital, Khon Kaen; and Faculty of Medicine (Srinagarind Hospital), Khon Kaen University, Khon Kaen, Thailand (approval no. HE541113). Subjects or their representatives provided written informed consent to be included within this study.

## 3. Results

Of the 327 patients with staphylococcal infection identified by the hospital microbiology laboratories in the original study, 74 patients were excluded from the final analysis. Five patients were excluded due to age < 18, and 69 patients were excluded because their specimens for culture were obtained > 24 h after hospital admission (Figure 1). A total of 253 patients met the inclusion criteria for this analysis. Of the 239 subjects whose bacterial isolates were obtained for genetic analysis (samples for 14 of the patients were discarded in error), 189/239 (79%) were confirmed to be *S. aureus* and 50/239 (21%) were reclassified as *S. argenteus*.

Patient characteristics are shown in Table 1. The median age was 56 years (IQR, 43–66 years), and 85 (34%) were female. No patients had 0 or 1 SIRS criteria, in accordance with the inclusion criteria; 93 (37%) had 2 SIRS criteria, 102 (40%) had 3 SIRS criteria, and 58 (23%) had 4 SIRS criteria. Of all 253 patients, 66 (26%) had a qSOFA score of 0, 118 (7%) had a qSOFA score of 1, 62 (25%) had a qSOFA score of 2, and 7 (3%) had a qSOFA score of 3.

Twenty-three (9%) patients died by 28 days. The proportion of patients who died increased linearly with qSOFA score (Figure 2 and Figure 3). Eight of 184 (4%) patients with qSOFA < 2 died compared to fifteen of 69 (22%) of patients with qSOFA ≥ 2.

A baseline risk model for mortality was developed using age, sex, cardiac disease, and liver disease. The AUROC for the baseline risk model alone was 0.62 (95% CI, 0.49–0.75). To quantify predictive validity, the qSOFA score was added to the baseline risk model to determine excess mortality above and beyond the baseline risk model. The addition of qSOFA score to the baseline risk model significantly increased the AUROC to 0.80 (95% CI, 0.70–0.89; *p* < 0.001 for difference, Figure 4).

*S. argenteus* is a relatively new member of a *S. aureus*-related complex [21]. The pathogenicity of this species in comparison to *S. aureus* is still being evaluated. Therefore, a sensitivity analysis was performed on patients whose cultures were confirmed to be *S. aureus* (*n* = 189) using molecular methods (Table 2). Of these patients, 47 (25%) had a qSOFA score of 0, 82 (44%) had a qSOFA score of 1, 54 (29%) had a qSOFA score of 2, and 6 (3%) had a qSOFA score of 3. Seventeen (9%) of the patients infected with *S. aureus* died by 28 days. Six of 129 (6%) patients with qSOFA < 2 died compared to 11 of 60 (18%) of patients with qSOFA ≥ 2.

Of patients with *S. aureus* infection, the AUROC for the baseline risk model alone was 0.67 (95% CI, 0.53–0.81). The addition of qSOFA score to the baseline risk model significantly increased the AUROC to 0.78 (95% CI, 0.68–0.89; *p* = 0.04; Figure 5).

## 4. Discussion

In this study of hospitalized adults in northeast Thailand with documented staphylococcal infection and at least two SIRS criteria, the qSOFA score demonstrated good predictive validity for sepsis. These findings suggest that the qSOFA score could be of utility to clinicians in LMICs to predict patients at risk of having sepsis. The ability to accurately identify patients likely to have sepsis in LMICs could help clinicians triage patients more effectively and allocate resources more efficiently.

As the qSOFA score was derived exclusively with data from HICs, it is important to ensure that it has generalizability to other populations if it is to be used as a decision-making tool in those clinical settings. Our findings add data in support of the use of the qSOFA score in this middle-income country setting as an effective tool to identify, from among patients with suspected infection, those who are likely to be septic. The findings of this study are concordant with other retrospective studies examining the utility of qSOFA for predicting sepsis in LMIC cohorts [14,22]. Although together these emerging data suggest that the value of qSOFA extends to LMICs, given the diversity in etiologies of sepsis, host characteristics, and resources available for diagnosis and treatment in LMICs, it will be important to conduct additional research to confirm these findings [23].

This is one of the first studies to specifically evaluate the predictive validity of qSOFA in patients with culture-proven coagulase-positive staphylococcal infection. A recent study by Minejima et al. examined patients with *S. aureus* bacteremia and found a significantly higher AUROC for 30-day mortality with qSOFA compared to SIRS in patients with documented *S. aureus* bacteremia in a HIC [24]. This is important, as *S. aureus* is an organism that is both a common cause of sepsis and has been shown to have increased risk of mortality compared to other pathogens [3,4,5,6]. As host inflammatory responses may be pathogen-specific, our focus on staphylococcal infections avoids potential heterogeneity confounding the analysis [25,26].

One potential advantage of the qSOFA score, as compared to other scoring systems also designed to predict the presence of sepsis, is that it requires no laboratory data. Thus, it can be completed quickly at the bedside and repeated as frequently as necessary. This attribute makes it an attractive scoring system in regions with limited resources.

Strengths of the study include enrollment of patients across multiple sites, cohort size, and limiting the scope of investigation to patients with only *S. aureus* or *S. argenteus* infection. There are also several limitations. As fulfilling two or more SIRS criteria was a required inclusion criterion, we are unable to directly compare the predictive validity of the SIRS criteria and qSOFA score. Moreover, the study may have limited generalizability to less ill patients, as those with only 0 or 1 SIRS criteria were excluded from the cohort. An additional limitation is that the recorded vital signs from which the qSOFA score was calculated were the most abnormal values within 48 h of culture collection. As a result, the predictive validity of the analyses may be favorably biased towards qSOFA, as compared to a qSOFA score calculation based on vital signs at the time of initial presentation to the hospital. The focus on predictive validity of qSOFA in staphylococcal infection may not reflect the predictive validity of qSOFA for other etiologies of infection and sepsis. Finally, this study was limited to adult patients, and thus does not capture neonatal or pediatric sepsis, which are significant causes of mortality in LMICs [27].

## 5. Conclusions

Within this cohort of adult patients with community-onset staphylococcal infection and at least 2 SIRS criteria in Thailand, the qSOFA score demonstrated good predictive validity for sepsis. These findings add to the literature in support of the qSOFA score as an effective tool for the identification of patients with sepsis in low-and-middle income countries.

## Figures and Tables

**Figure 1 jcm-08-01908-f001:**
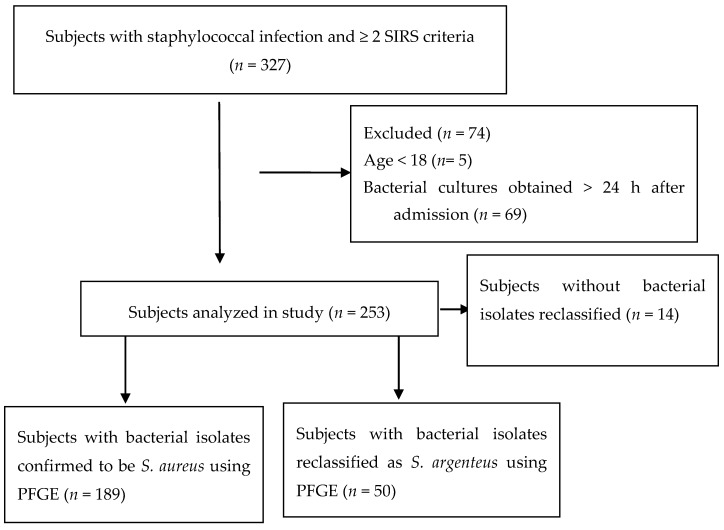
Number of patients included in the secondary analysis. Abbreviations: PFGE = Pulsed field gel electrophoresis.

**Figure 2 jcm-08-01908-f002:**
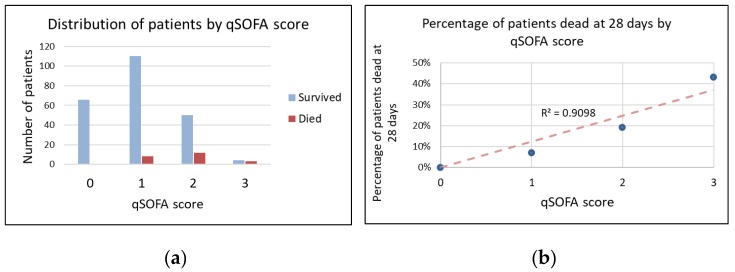
Distribution of patient mortality based on quick sequential organ failure assessment (qSOFA) score. (**a**): Distribution of patients by qSOFA score; (**b**): Percentage of patients dead at 28 days by qSOFA score.

**Figure 3 jcm-08-01908-f003:**
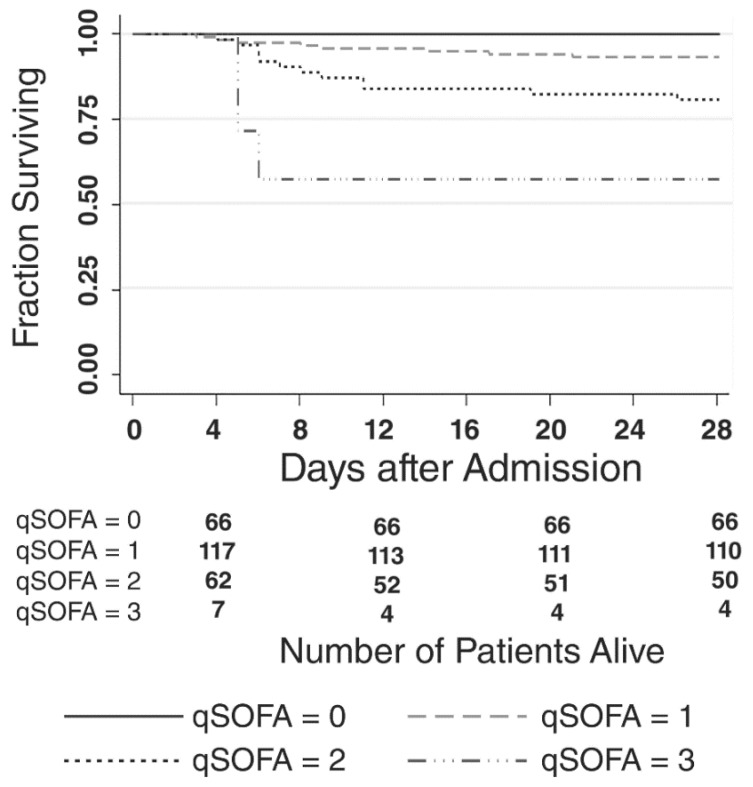
Survival by quick sequential organ failure assessment score at time of admission. The graph demonstrates the fraction of patients surviving over 28 days based on qSOFA. score. The table below indicates the number of patients alive within each qSOFA cohort at days 4, 12, 20, and 28.

**Figure 4 jcm-08-01908-f004:**
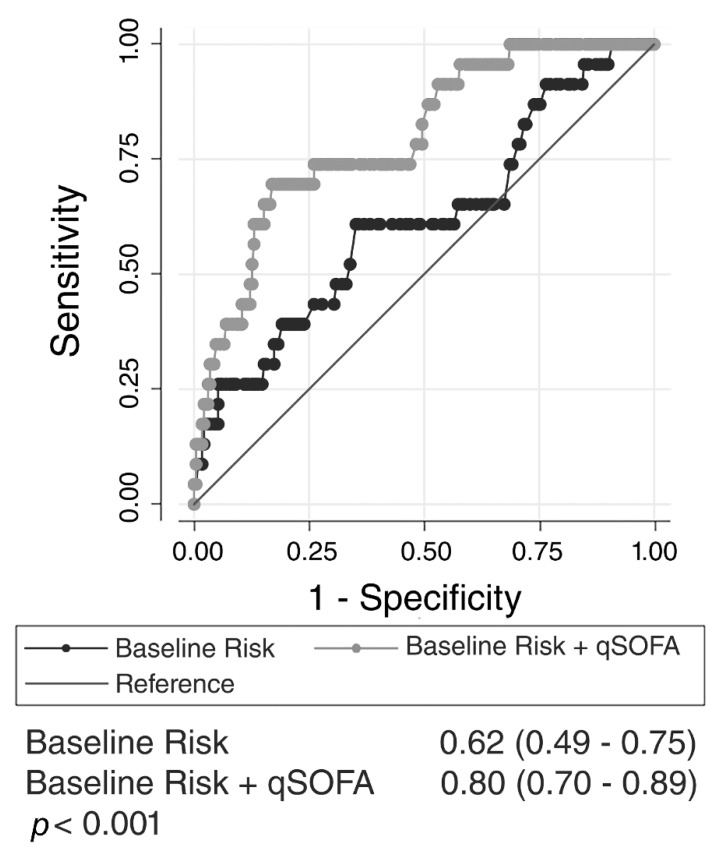
Receiver operating characteristic curves for baseline risk compared to baseline risk added to quick sequential organ failure assessment (qSOFA) score for all patients with documented staphylococcal infection. Baseline risk model includes age, sex, and co-morbidities associated with the outcome.

**Figure 5 jcm-08-01908-f005:**
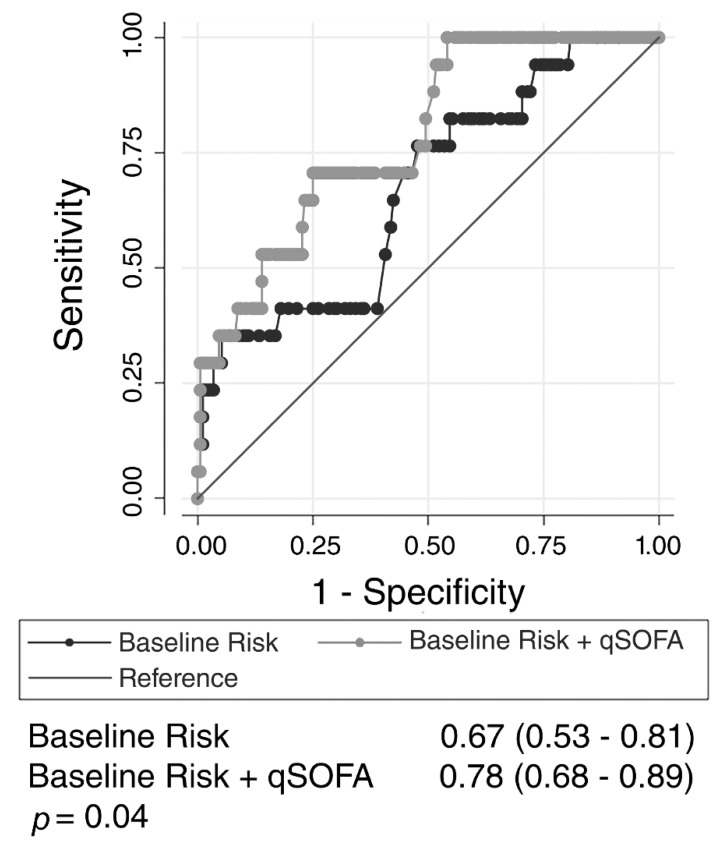
Receiver operating characteristic curves for patients with *Staphylococcus aureus*. Baseline risk compared to baseline risk added to quick sequential organ failure assessment (qSOFA) score. Baseline risk model includes age, sex, and co-morbidities associated with the outcome.

**Table 1 jcm-08-01908-t001:** Patient characteristics.

Variables	All Patients	Alive at 28 Days	Dead at 28 Days
	*n* = 253	*n* = 230	*n* = 23
Age (years), median (IQR)	56 (43–66)	56 (42–66)	56 (46–65)
Male, *n* (%)	168 (66%)	152 (66%)	16 (70%)
Systolic blood pressure (mmHg), median (IQR)	102 (95–110)	103 (96–110)	93 (80–104)
Glasgow Coma Scale ≤ 14, *n* (%)	14 (5%)	8 (3%)	6 (25%)
Respiratory rate > 20, *n* (%)	130 (51%)	111 (48%)	19 (79%)
Liver disease present, *n* (%)	9 (4%)	5 (2%)	4 (17%)
Cardiac disease present, *n* (%)	12 (5%)	11 (5%)	1 (4%)
SIRS criteria, *n* (%)			
0	0 (0%)	0 (0%)	0 (0%)
1	0 (0%)	0 (0%)	0 (0%)
2	93 (37%)	88 (38%)	5 (22%)
3	102 (40%)	95 (41%)	7 (30%)
4	58 (23%)	47 (20%)	11 (48%)
qSOFA score, *n* (%)			
0	66 (26%)	66 (29%)	0 (0%)
1	118 (47%)	110 (48%)	8 (38%)
2	62 (25%)	50 (22%)	12 (50%)
3	7 (3%)	4 (2%)	3 (13%)
qSOFA score ≥ 2, *n* (%)	69 (27%)	54 (23%)	15 (65%)

Abbreviations: IQR, interquartile range; SIRS, systemic inflammatory response syndrome; qSOFA, quick sequential organ failure assessment.

**Table 2 jcm-08-01908-t002:** Characteristics of patients with *Staphylococcus aureus* infection.

Variables	All Patients	Alive at 28 Days	Dead at 28 Days
	*n* = 189	*n* = 172	*n* = 17
SIRS criteria, *n* (%)			
0	0 (0%)	0 (0%)	0 (0%)
1	0 (0%)	0 (0%)	0 (0%)
2	66 (37%)	63 (37%)	3 (18%)
3	72 (38%)	66 (38%)	6 (35%)
4	51 (27%)	43 (25%)	8 (47%)
qSOFA score, *n* (%)			
0	47 (25%)	47 (27%)	0 (0%)
1	82 (44%)	76 (44%)	6 (35%)
2	54 (29%)	45 (26%)	9 (53%)
3	6 (3%)	4 (2%)	2 (12%)
qSOFA score ≥ 2, *n* (%)	60 (32%)	49 (28%)	11 (65%)

Abbreviations: SIRS, systemic inflammatory response syndrome; qSOFA, quick sequential organ failure assessment.

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
