# Peer review of "Predictive Validity of the qSOFA Score for Sepsis in Adults with Community-Onset Staphylococcal Infection in Thailand"

_jcm, 2019, doi:10.3390/jcm8111908_

Round 1

Reviewer 1 Report

The manuscript “Predictive Validity of the qSOFA Score for Sepsis in Adults with Community-Onset Staphylococcal Infection in Thailand” describes the secondary analysis of a prospective multicenter cohort study in Thailand to evaluate the predictive validity of the qSOFA score for sepsis within a low-and middle-income country population with culture-proven staphylococcal infection.

The findings of the study are novel; no literature on association between qSOFA and PFGE culture-proven staphylococcal infection is yet available. The study was well-designed and executed. Efforts of the authors in carrying out the study are commendable.

Below are my comments for the authors:

Introduction:

The introduction is concise and well-written.

Methodology:

1.     The qSOFA score (which only consists of 3 clinical parameters) can be easily assessed at the time of initial presentation in hospital. Can the authors clarify, why they chose the most out-of-range qSOFA scores (taken within 48hours of hospitalization) for analysis, instead of the 1st set of qSOFA scores upon patient arrival to the hospital?

2.     qSOFA is recommended to be used for outside ICU setting. However, it was stated that patients admitted to acute and intensive care wards were included for analysis. Can the authors explain the rationale for the inclusion, and state the exact number of patients who originated from ICU in this cohort?

3.     Usage of the criteria: respiratory rate >20/min (equivalent to 1 qSOFA score) in those who were ventilated needs further justification.

4.     What was the sampling method used in this prospective cohort study? Was the sampling randomized or was it carried out consecutively?

5.     Ethical statement: How was informed consent obtained during the study? This process shall be stated in the manuscript.

Results

The results were well-presented.

Minor error: The number of subjects for S. aureus and S. argenteus in figure 1 are missing.

Discussion

Discussion was well-written.

The authors did not analyse the performance of SIRS compare to qSOFA. Hence, the findings of this study cannot be stated to be in concordance with those demonstrated by Mineja et al.

Author Response

Thank you for the opportunity to revise our manuscript, “Predictive Validity of the qSOFA Score for Sepsis in Adults with Community-Onset Staphylococcal Infection in Thailand” (JCM-628030). We are grateful to the reviewers for their thoughtful comments and suggestions. We have modified the manuscript, and provide point-by-point responses to each comment below. The reviewers’ comments are in bold and our responses are beneath. Specific changes are noted in tracked changes in the marked-up version of the original manuscript.

The qSOFA score (which only consists of 3 clinical parameters) can be easily assessed at the time of initial presentation in hospital. Can the authors clarify, why they chose the most out-of-range qSOFA scores (taken within 48hours of hospitalization) for analysis, instead of the 1st set of qSOFA scores upon patient arrival to the hospital?

This is an excellent point and we agree that this is a limitation of our study. As this was a secondary analysis, we were limited to the data collection of the original study. The only data points collected in the repository were the most out-of-range SIRS criteria within the first 48 hours of hospitalization. As a result of this, our analysis is also limited to the most abnormal qSOFA scores within the first 48 hours of hospitalization as opposed to the initial qSOFA scores that would have been taken upon arrival to the hospital. This limitation has been acknowledged within the discussion section.

qSOFA is recommended to be used for outside ICU setting. However, it was stated that patients admitted to acute and intensive care wards were included for analysis. Can the authors explain the rationale for the inclusion, and state the exact number of patients who originated from ICU in this cohort?

We agree that qSOFA is recommended for use outside of the ICU setting and is a less robust system in this setting than SOFA. Given that the SOFA and qSOFA scoring system were developed from data based on cohorts from developed countries we felt that it would be more appropriate in our study to include all patients included to the hospital, including the ICU. Moreover, given the advantage of qSOFA’s ease of obtainment in resource-limited settings, we felt it would be advantageous to assess its utility in ICU’s in less developed settings where all the lab values needed to calculate a SOFA score may not be as readily available. Overall, of the 253 patients included in our study only 11 were initially admitted to the ICU.

Usage of the criteria: respiratory rate >20/min (equivalent to 1 qSOFA score) in those who were ventilated needs further justification.

Optimally we would have had the exact respiratory rates collected for each individual patient. The original study design was created to capture sepsis based on SIRS criteria. As a result, patients were given one point for the SIRS criteria of respiratory rate if their respiratory rate was above 20 and the actual value was not recorded in our data repository. Thus, to meet the qSOFA criteria of RR ≥ 22 we had to substitute those who had a RR > 20.

What was the sampling method used in this prospective cohort study? Was the sampling randomized or was it carried out consecutively?

The sampling method used within this prospective cohort study was carried out consecutively.

Ethical statement: How was informed consent obtained during the study? This process shall be stated in the manuscript.

Thank you for this clarification. This has been updated under the Ethics subsection in the Methods.

Minor error: The number of subjects for S. aureus and S. argenteus in figure 1 are missing.

Thank you for discovering this – the number of subjects appeared to be formatted out due to the size of the text boxes. This has been fixed.

The authors did not analyse the performance of SIRS compare to qSOFA. Hence, the findings of this study cannot be stated to be in concordance with those demonstrated by Mineja et al.

Thank you for this important and clarifying statement. We agree with the reviewer and have removed the statement of concordance.

Reviewer 2 Report

The manuscript is generally well-written. However, it can be improved according to the following pointers:

General comments:

There is a general lack of flow in the results section. Preferably, the results section can be split into several sub sections to facilitate reader's understanding.

Specific comments:

How was the baseline risk model calculated? Right now, the description is not detailed enough other than stating that 'logistic regression was used to identify co-morbidities associated with mortality at 28 days'. This is merely a description of feature selection for the actual model, the details of which are currently lacking.  Page 3, lines 103 and 104: 'autoimmune disease' is repeated. Page 3, line 3: '311 of the 327 patients'. Why was PFGE typing performed on a subset of the total patients? Also, on page 4, line 156-157, out of 253 patients, 239 were typed for bacterial species. Please explain the rationale behind these decisions. Page 2, line 88: The authors mention that 'Included patients were >= 14 years old'. However, on page 3, line 119, it is mentioned that 'For this analysis, patients were excluded if they were <18 years old'. Please clarify these potentially conflicting statements. Page 4, lines 154-156: It appears that the inclusion criteria for enrollment and for statistical analysis are different (age: 14 years vs. 18, time of culture: 48 hours vs 24). Please explain why a different inclusion criteria was employed for the statistical analyses. Page 6, line 184, in relation to figure 2: 'increased linearly'. Instead of the current plot, a better way to represent the data could be a bivariate scatterplot of qSOFA scores and percent death fitted with a linear model to demonstrate if indeed the two scale linearly. Figure 3: Please add a detailed legend about the figure. Particularly the table. What do the columns mean? Figures 4 and 5: Please state explicitly in the legend to figure 4 if the analyses were performed with both cases of S.aureus and S.argenteus. Assuming this is the case, the AUROC values are not significantly improved by filtering out cases of S.argenteus infections. As such, my recommendation would be to move figure 4 to supplementary information. That way, the focus is always on the cases of S.aureus, which has been so explicitly stated right from the begining of the manuscript.

Author Response

Thank you for the opportunity to revise our manuscript, “Predictive Validity of the qSOFA Score for Sepsis in Adults with Community-Onset Staphylococcal Infection in Thailand” (JCM-628030). We are grateful to the reviewers for their thoughtful comments and suggestions. We have modified the manuscript, and provide point-by-point responses to each comment below. The reviewers’ comments are in bold and our responses are beneath. Specific changes are noted in tracked changes in the marked-up version of the original manuscript.

How was the baseline risk model calculated? Right now, the description is not detailed enough other than stating that 'logistic regression was used to identify co-morbidities associated with mortality at 28 days'. This is merely a description of feature selection for the actual model, the details of which are currently lacking.

We thank the reviewer for this clarification. We have added additional details indicating that our baseline risk model of mortality was created using generalized estimating equations with a panel-data model using binomial family, logit link, and robust standard errors.

 Page 3, lines 103 and 104: 'autoimmune disease' is repeated.

Thank you for identifying this redundancy which has now been removed.; this has been fixed.

Page 3, line 3: '311 of the 327 patients'.

We have modified this portion for further clarity, we thank the reviewer for bringing this to our attention.

Why was PFGE typing performed on a subset of the total patients? Also, on page 4, line 156-157, out of 253 patients, 239 were typed for bacterial species. Please explain the rationale behind these decisions.

We appreciate this clarification. The bacterial isolates for 14 of the patients from the original study were discarded in error and thus were not available for PFGE typing for those particular patients which is why only 239 patients were typed for bacterial species. We have updated the manuscript to reflect this information.

Page 2, line 88: The authors mention that 'Included patients were >= 14 years old'. However, on page 3, line 119, it is mentioned that 'For this analysis, patients were excluded if they were <18 years old'. Please clarify these potentially conflicting statements.

Our apologies for this confusion. The original inclusion criteria included patients greater than or equal to fourteen years of age because in Thailand that is the age cut-off for admittance to the acute care ward. In order to avoid inclusion of pediatric sepsis, we limited our secondary analysis of the original study to patients greater than or equal to 18. We have rephrased these two lines for added clarity.

Page 4, lines 154-156: It appears that the inclusion criteria for enrollment and for statistical analysis are different (age: 14 years vs. 18, time of culture: 48 hours vs 24). Please explain why a different inclusion criteria was employed for the statistical analyses.

As discussed above, the age cut-off for our secondary analysis was made to be 18 and thus patients between the ages of 14-17 that were included in the original study were excluded from our analysis. Similarly, we excluded patients whose cultures were positive > 24 hours after admission (which excluded 69 patients from the original analysis) so as to focus on community-acquired infections and avoid possible hospital-acquired infections.

Page 6, line 184, in relation to figure 2: 'increased linearly'. Instead of the current plot, a better way to represent the data could be a bivariate scatterplot of qSOFA scores and percent death fitted with a linear model to demonstrate if indeed the two scale linearly.

We greatly appreciate and agree with this suggestion by the reviewer. We have modified our data to show that the percent death indeed does appear to scale linearly with the increase in qSOFA score.

Figure 3: Please add a detailed legend about the figure. Particularly the table. What do the columns mean?

Additional information has been added to provide clarity on the information contained within the figure and the columns as well.

Figures 4 and 5: Please state explicitly in the legend to figure 4 if the analyses were performed with both cases of S.aureus and S.argenteus. Assuming this is the case, the AUROC values are not significantly improved by filtering out cases of S.argenteus infections. As such, my recommendation would be to move figure 4 to supplementary information. That way, the focus is always on the cases of S.aureus, which has been so explicitly stated right from the beginning of the manuscript.

We thank the reviewer for this suggestion. Figure 4 was conducted with all patients with staphylococcal infection – S. aureus and argenteus included. We have updated the header of the image to make this clearer to the readers. We agree that that the suggestion provided to move figure 4 to supplementary information is very reasonable, we have opted to leave the figures as is given that there is thus far limited data to suggest that the infectious profile of S. aureus and argenteus are significantly different and would like to focus on staphylococcal infections in total.